# Spatio-Temporal Dynamic of Malaria Incidence: A Comparison of Two Ecological Zones in Mali

**DOI:** 10.3390/ijerph17134698

**Published:** 2020-06-30

**Authors:** François Freddy Ateba, Issaka Sagara, Nafomon Sogoba, Mahamoudou Touré, Drissa Konaté, Sory Ibrahim Diawara, Séidina Aboubacar Samba Diakité, Ayouba Diarra, Mamadou D. Coulibaly, Mathias Dolo, Amagana Dolo, Aissata Sacko, Sidibe M’baye Thiam, Aliou Sissako, Lansana Sangaré, Mahamadou Diakité, Ousmane A. Koita, Mady Cissoko, Sékou Fantamady Traore, Peter John Winch, Manuel Febrero-Bande, Jeffrey G. Shaffer, Donald J. Krogtad, Hannah Catherine Marker, Seydou Doumbia, Jean Gaudart

**Affiliations:** 1Malaria Research and Training Center, Faculty of Medicine, Pharmacy and Dentistry, University of Sciences, Techniques and Technologies of Bamako, Bamako BP 1805, Mali; freddy.francois.ateba@gmail.com (F.F.A.); isagara@icermali.org (I.S.); nafomon@icermali.org (N.S.); mtoure@icermali.org (M.T.); dkonate@icermali.org (D.K.); sdiawara@icermali.org (S.I.D.); sdiakite@icermali.org (S.A.S.D.); ayouba.diarra@icermali.org (A.D.); mdcoulibaly@icermali.org (M.D.C.); mathiasdolo442@yahoo.com (M.D.); adolo@icermali.org (A.D.); sibe_t@icermali.org (S.M.T.); mdiakite@icermali.org (M.D.); madycissoko@ymail.com (M.C.); cheick@icermali.org (S.F.T.); 2Department of Mathematics, University of Quebec at Montreal (UQAM), Montréal, QC H2X 3Y7, Canada; 3Department of Public Health Education and Research, Faculty of Medicine and Odonto-Stomatology, University of Sciences, Techniques and Technologies of Bamako, Bamako BP 1805, Mali; aissatasacko1989@gmail.com; 4Laboratory of Applied Molecular Biology (LBMA), Science and Technologies Faculty (FST), University of Sciences, Techniques and Technologies of Bamako, Bamako BP 1805, Mali; drsissakoaliou@yahoo.fr (A.S.); lansana.sangare@gmail.com (L.S.); oakoita@yahoo.com (O.A.K.); 5APHM, INSERM, IRD, SESSTIM, Hop Timone, BioSTIC, Biostatistic & ICT, Aix Marseille Université, 13005 Marseille, France; 6Johns Hopkins Bloomberg School of Public Health, Johns Hopkins University, Baltimore, MD 21205, USA; pwinch@jhu.edu (P.J.W.); hannah.marker@jhu.edu (H.C.M.); 7Department of Statistics, Mathematical Analysis and Optimization, University of Santiago de Compostela, 15782 Santiago de Compostela, Spain; manuel.febrero@usc.es; 8Department of Global Biostatistics and Data Science, School of Public Health and Tropical Medicine, Tulane University, New Orleans, Louisiana, United States of America, 1440 Canal Street New Orleans, LA 70112, USA; jshaffer@tulane.edu (J.G.S.); krogstad@tulane.edu (D.J.K.)

**Keywords:** malaria, generalized additive models, geo-epidemiology, lag, normalized difference vegetation index, principal components analysis, passive case detection, plasmodium falciparum

## Abstract

Malaria transmission largely depends on environmental, climatic, and hydrological conditions. In Mali, malaria epidemiological patterns are nested within three ecological zones. This study aimed at assessing the relationship between those conditions and the incidence of malaria in Dangassa and Koila, Mali. Malaria data was collected through passive case detection at community health facilities of each study site from June 2015 to January 2017. Climate and environmental data were obtained over the same time period from the Goddard Earth Sciences (Giovanni) platform and hydrological data from Mali hydraulic services. A generalized additive model was used to determine the lagged time between each principal component analysis derived component and the incidence of malaria cases, and also used to analyze the relationship between malaria and the lagged components in a multivariate approach. Malaria transmission patterns were bimodal at both sites, but peak and lull periods were longer lasting for Koila study site. Temperatures were associated with malaria incidence in both sites. In Dangassa, the wind speed (*p* = 0.005) and river heights (*p* = 0.010) contributed to increasing malaria incidence, in contrast to Koila, where it was humidity (*p* < 0.001) and vegetation (*p* = 0.004). The relationships between environmental factors and malaria incidence differed between the two settings, implying different malaria dynamics and adjustments in the conception and plan of interventions.

## 1. Introduction

In Mali, despite concerted efforts from national and international partners to scale up effective malaria control interventions, malaria incidence remains a public health concern [1]. In Malian health facilities, the World Health Organization (WHO) has reported 2.28 million cases and 12,425 deaths due to malaria in 2017 and highlighted an increase of more than 100,000 cases between 2016 and 2017 [2].

In 2018, the cumulative annual number of suspected malaria cases in Mali was 2,834,142. Among these suspected cases, approximately 98.00% were tested, out of which 1,757,292 cases were confirmed (60.28%). Among the confirmed cases, approximately 34.00% occurred in children under five years of age. Compared to 2017, the number of confirmed cases increased by 16.50%. A case-fatality rate of 0.65% was recorded in 2018, and a slight decrease (less than 1%) in the case-fatality rate was observed in 2017 [3].

Malaria control in Mali is primarily based on improving early access to diagnostics and treatment, which includes: (i) intermittent preventive treatment in pregnancy (IPTp, launched in 2003); (ii) free distribution of insecticide-treated nets or long lasting insecticidal Net (ITNs/LLINs, launched in 2005); (iii) use of artemisinin combination therapy (ACT, launched in 2006 and a test-and-treat policy, implemented in 2010); and (iv) indoor residual spraying (IRS, launched in two districts in 2008). According to the Malaria Indicator Survey (MIS, Mali 2017), 92.20% of households had at least one long lasting insecticidal Net (LLIN), 39.00% had at least one LLIN for two people, 68.00% of people of all ages were sleeping under LLINs, 75.00% of children under five years slept under LLINs and 78.00% of pregnant women slept under LLINs [4].

Like any vector-borne diseases, malaria transmission is largely dependent on environmental, meteorological and hydrological conditions such as temperature, rainfall, vegetation, and variations in river heights across the considered regions [5,6,7]. In Mali, malaria epidemiological patterns vary from a sporadic or epidemic transmission to low and high transmission. Those patterns are nested within three ecological zones: (i) a Sahelian area; (ii) a Sudan Savanna area; and (iii) an irrigated area [8,9]. Several studies in West Africa, and particularly in Mali, have highlighted the complex relationship between socioeconomic, hydrological, climatic, anthropological factors and malaria incidence. Such heterogeneity may bring important variations in malaria transmission patterns within the country [10,11,12].

In a changing environment and a limited-resources setting similar to Mali, malaria geo-epidemiology will help understand the spatial and temporal dynamics of malaria. This will help deploy, monitor and evaluate, with a strategic adaptive approach, a number of sustainable control and elimination interventions [13]. The aim of this study was to assess and contrast the key environmental factors involved in malaria transmission dynamics in two different ecological settings of Mali.

## 2. Materials and Methods

### 2.1. Materials

#### 2.1.1. Study Areas

This study was carried out in two remote villages situated in Mali’s Kati and Markala districts. The first of these sites, Dangassa, is located in the Sudan Savanna area, about 115 km from Bamako, in the district of Kati, Koulikoro region (See Figure 1). The main village is approximately 4 km from the Niger River and its hamlets lay along the river. Dangassa had an estimated population of 6200 inhabitants in 2012 [14] and is located at an altitude 350 m. During the dry season, they are microhabitats for mosquito breeding, and the footprints of cattle emerge and leave natural pools. Manmade pools are also left near the river due to gold mining. Gold mining is one of the main activities in the village, and as a consequence, it may be related to malaria transmission in the village [15,16]. The average annual temperature is 27.50 °C, with an average rainfall of 855 mm per year [17]. Community health centers in Dangassa cover 16 surrounding hamlets. Since 2008, specific interventions targeting malaria control in vulnerable population groups (children < 5 years old, pregnant women) have been made available in the village: LLINs, rapid diagnostic tests (RDTs), ACTs for treatment of acute symptomatic *P. falciparum* infection, and IPTp using a combined sulfadoxine and pyrimethamine regimen (SP).

The second site, Koila, is located in an irrigated Sahelian area of the Dioro in Mali’s Markala District and Ségou Region (See Figure 1), located 399 km from Bamako, at an altitude of 365 m. This locality had an estimated population of 6100 inhabitants in 2012 [14]. The site has an average annual temperature of 28.70 °C and average annual rainfall of 492.9 mm [17]. Koila is characterized by an irrigation scheme from the Markala Dam on the Niger River, which holds back water flowing downstream from Ségou toward Mopti. Water released in August regularly floods the area, allowing the cultivation of rice until December. Later, a second set of gates releases water, allowing growth of a second rice crop in April, harvested in May. This situation results in virtually year-round malaria transmission, driven by irrigation from August to May and by the rainy season from July to November. The main agricultural crop in the region is rice. Intensive malaria control interventions in the area began in 2006 and include: universal use of LLINs, ACT for treatment of acute symptomatic *P. falciparum* infection, active case detection by RDT, treatment by village health workers, IPTp and universal health coverage with free treatment in local health facilities. From November 2007 to 2014, the Millennium Villages Project (MVP) was launched in the 39 villages of Dioro District, including Koila. During its implementation, the MVP provided intensive malaria control to Dioro and other communities in the Ségou Region of Mali. Those interventions included ACT and pro-active replacement of worn or torn bed nets. As a result, malaria transmission and morbidity have fallen to unprecedented levels [18].

#### 2.1.2. Data Source

##### Study Population and Data Collection

The data originated from the registers of community health centers of each study site from 22 June 2015 to 6 January 2017. Passive case detection (PCD) was carried out year-round in both the dry season (December to June) and the rainy season, which corresponds to the high malaria transmission period (July to October/November). Inhabitants of the villages were encouraged to present to the community health center if experiencing fever or illness, where they were examined by the study’s physicians. The village community health workers were encouraged to refer all inhabitants with fever to the study physicians for physical examination and malaria diagnosis. During the clinical examination, all patients belonging to the age range 0 to 85 with fever (temperature ≥ 37.50 °C) or a history of fever in the last 48 h were tested for malaria using an RDT. Blood smears were collected for parasite species identification by microscopy.

Malaria episode was defined as fever or history of fever with positive RDT. For any confirmed case of malaria, malaria treatment was freely provided by the community health center staff according to the recommendations of the Ministry of Health. Parents were encouraged to return with their children 4 to 6 days after the start of treatment to ensure clearance of parasites and to rule out the possibility of drug resistance. Periodic review and quality control were performed at community health centers. At the time of the most recent study census, all residents, including children and adults, received an identification card (ID) with their unique identification number and photo. Parents were asked to present their children’s ID cards at each visit to the village health center. A computerized database of the study population was constructed and updated quarterly with census data (e.g., newborn babies and migration/immigration).

##### Hydrological Data

Daily measurements of the height of the Niger River near the study sites were collected by the National Directorate of Hydraulics of Mali throughout the study period. These are measurements of average daily readings made by the sensors along the Bamako main river arm, the nearest hydrologic station to Dangassa (60 km), and the Kirango-Aval (Diamarabougou, Markala) sensors from the nearest station to Koila (45 km).

##### Climate and Environmental Data

Meteorological data used for each of the sites as part of this analysis were extracted from the Earth Observing System Data and Information System (EOSDIS, https://urs.earthdata.nasa.gov) from NASA from 22 June 2015 through 31 December 2016. The same mode of collection was performed for the Normalized Difference Vegetation Index (NDVI) over the same period. The data extracted for these analyses were: (1) precipitation (mm per day) with a spatial resolution of 0.25°; (2) the minimum, average, and maximum air temperature (°C) with a resolution of 0.5 × 0.625° [daily]; (3) the daily relative humidity (%); (4) the air temperatures at the ground surface with a spatial resolution of 1. (5) the daily wind speed (m/s) with a spatial resolution of 0.25°; and (6) the monthly index of vegetation (NDVI) with a spatial resolution of 0.05°.

### 2.2. Statistical Methods

To assess the relationship between environmental, meteorological, and hydrological components and the incidence of malaria at the different sites, the same analytical approaches were applied for the Dangassa and Koila study sites. Epidemiological data (number of malaria cases), hydrological data (river height), and meteorological and environmental data (temperature, humidity, vegetation index, wind speed, rainfall) were aggregated from a daily to weekly basis, except for NDVI data, which was captured on a monthly basis (EOSDIS information system). New variables were generated on the basis of the values of weekly primitive variables. Summary measures included weekly minimum values, maximal values, mean values, interquartile intervals of observed weekly values, cumulative values for precipitation and weekly number of rain events based on weekly rainfall values [10,12].

While the environmental variables were highly collinear and numerous, principal component analyses (PCA) were used to reduce collinearities and dimension. Two separate PCAs were carried out on hydrological variables (height of the Niger River) and on meteorological and vegetation variables [10,12]. The approach allowed us to provide environmental synthetic indices. To quantify the impact of the environmental synthetic indices, we used a generalized additive model (GAM) approach [19,20,21,22,23,24,25,26]. GAM modeling has been used to model epidemiological, meteorological, and public health phenomena. Several studies on the relationship between malaria and the environment have used this modeling approach because of its flexibility, which allows modelling of nonlinear relationships. We used the GAM modelling approach in conjunction with spline smoothing to model the non-linear relationship between malaria incidence and the environmental synthetic indices issued from the PCA [10,11,12,27,28].

To investigate the immediate and delayed effects of meteorological and hydrological components on the incidence of malaria, a univariate GAM model was applied to each of the environmental synthetic indices. Population data were log-transformed to provide estimations of the standardized incidence ratios (SIR). A logarithmic (canonical) link function using a quasi-Poisson model was used to account for overly dispersed data.

Predictor variable sets were made up of each of the single environmental synthetic indices derived from each of the principal component analyses. Associations were assessed for different time lags and selected using the unbiased risk estimator (UBRE) and deviance criteria [29]. In an effort to determine the associations between the occurrence of malaria cases and environment components, multivariate GAM modeling was applied to all lagged covariates. All statistical analysis were performed by using R 3.5. [30] using the following packages: (1) FactoMineR [31]; (2) mgcv [29]; (3) lattice [32]; (4) forecast [33]; (5) factoextra [34]; (6) corrplot [35]; and (7) imputeTS, [36]. Disease maps were generated using the ArcGIS application (Environmental Systems Research Institute, Redlands, CA, Version 10.3; [37]).

Research Data: The dataset of malaria cases aggregated on a weekly basis are available at the level of the ICEMR data management core (sdoumbi@icermali.org). The river data are available at the Mali hydraulic services. The meteorological and environmental data are free of access and available part of this analysis were extracted from the EOSDIS information system (https://urs.earthdata.nasa.gov).

## 3. Results

### 3.1. Time Series Overview

During the study period, symptomatic malaria cases diagnosed by the medical teams for Dangassa and Koila were 2014 and 713, respectively. The maximum number of cases was 49 per 1000 person-weeks in Koila and 70 per 1000 person-weeks in Dangassa. Malaria transmission was seasonal with bimodal periodicity for both sites. Malaria incidence time-series of both sites also revealed annual periodicities in the transmission dynamics. The transmission period lasted from July to March, with the highest transmission period lasting from July to October, in Dangassa (Figure 2). Koila revealed a transmission period that occurred from August to May, with the highest transmission period from July to December (Figure 3).

In the overall study period, the highest cumulative rainfall has been observed in Dangassa with 1366.03 mm, while less rainfall was observed in Koila (1138.59 mm). The highest maximum and minimum temperatures were recorded in Koila, with 44.70 °C and 29.84 °C, while Dangassa reached a maximum and minimum of 43.26 °C and 26.51 °C. The mean river height was the highest in Koila at 210.97 cm, while it was only 150.39 cm in Dangassa.

### 3.2. Principal Component Analysis

#### 3.2.1. Hydrological Components

For Dangassa and Koila, the combination of hydrological components has made it possible to identify two major components derived from the PCA, representing 99.90% and 100% of the inertia for both sites (see Figure 4C and Figure 5C). The two first components (Dhriv1 and Khriv1) representing 80.50% and 81.20% of the inertia, were composed mainly of higher river height while the two second components (Dhriv2 and Khriv2), representing 19.40% and 18.80% of the inertia, were constituted mainly of the variations in river height.

#### 3.2.2. Weather and Vegetation Components

The combination of meteorological and vegetation components has made it possible to identify three major components (derived from the PCA), accounting for 72.80% and 70.30% of total inertia for Dangassa and Koila, respectively (Figure 4A,B and Figure 5A,B). The composition of the PCA derived axis was variable across study sites. For the Dangassa group, the first components (Dmt1, 40.70% of the inertia) consisted mainly of humidity, vegetation, rain, and temperature, in opposition to the wind speed and the variation of temperatures. The second component (Dmt2, 24.70% of the inertia) was constituted mainly of temperature. The third component (Dmt3, 7.40% of the inertia) consisted of the wind speed and the variation of temperatures.

For the Koila study site, the first component (Kmt1, 34.80% of the inertia) consisted of humidity, rain, and minimum temperature, in opposition to the wind speed. The second component, (Kmt2, 27.50% of the inertia), consisted of the maximum temperature, opposed to vegetation indices (NDVI). The third component, (Kmt3, 8.00% of the inertia), consisted of the variations of all the meteorological components. 

### 3.3. Univariate GAM Modeling

For hydrological components, the univariate GAM modeling (Table 1) for the two sites revealed some differences in lag duration. For the first hydrological component made up of higher river height, the lag was shorter in Dangassa, (7 week lag, *p* = 0.002), which was less than the 11 week lag (*p* = 0.353) found for Koila. For the second hydrological component made up of the variations in river heights, the lag was shorter in Koila (5 week lag, *p* = 0.270) compared to the 8 week lag (*p* = 0.320) obtained in Dangassa (See Table 1).

For meteorological components, the first and main meteorological component were made up mainly of humidity in Dangassa and had the same composition in both sites (Table 1). The lag was shorter in Koila, with an 11 week lag (*p* = 0.001) compared to the 14 week lag (*p* < 0.001) in Dangassa. The second meteorological components, made up mainly of higher temperatures in both sites, showed a 14 week lag (*p* < 0.001) and a 5 week lag (*p* = 0.205) for Dangassa and Koila, respectively. The third meteorological component, which consisted mainly of the wind speed in Dangassa and the vegetation in Koila, showed a 3 week lag (*p* = 0.204) and 8 week lag (*p* = 0.531) for Dangassa and Koila, respectively (Table 1).

### 3.4. Multivariate GAM Modeling of Malaria Incidence

The purpose of the multivariate GAM analysis was to investigate the relationship between malaria incidence and all components (meteorological, environmental, and hydrological) issued from the principal component analysis, taking into account the lag times found in the previous univariate analysis. The multivariate GAM model revealed an explained deviance of 65.00% for the site of Dangassa and 48.20% for Koila (Table 1).

The first main synthetic meteorological component, mainly constituted by humidity, was not statistically significant in Dangassa (*p* = 0.540), in contrast to Koila, where the similar component was found to be significant (*p* < 0.001) and to have a linear relationship with malaria incidence (Figure 6A and Figure 7A). The second meteorological component was made up mainly of higher temperatures in both sites and was significantly associated with malaria incidence. In Dangassa, the relationship between the higher temperatures and malaria incidence was non-linear, showing that the incidence decreases with higher temperatures (*p* = 0.009; Figure 6B). In Koila, the relationship has been found to be quasi-linear and negative (*p* = 0.030; Figure 7B).

In Dangassa, the third meteorological component, which consisted mainly of the wind speed, was significantly associated to malaria incidence (*p* = 0.005). That component has a linear relationship with malaria incidence (Figure 6C), showing that less abrupt variations in wind speed decrease the incidence. In Koila, the third meteorological component was made up mainly of vegetation and was significantly associated with malaria incidence (*p* = 0.004), the relationship was quasi-linear (Figure 7C).

Neither of the two hydrological components were found to be significantly associated to malaria incidence in Koila (*p* = 0.660; Figure 6D). In contrast, in Dangassa, the hydrological component has been found to have a quasi-linear, positive, and significant relationship (*p* = 0.010; Figure 7D).

We noticed a shift in statistical significance for some covariates in both sites as we moved from the univariate to multivariate analysis. In Koila, the higher temperatures and the vegetation became significant (*p* = 0.004; *p* = 0.030 respectively). In Dangassa, the humidity became non-significant (*p* = 0.540) when the wind speed moved to significance (*p* = 0.005).

## 4. Discussion

Despite significant investments deployed by Sub-Saharan African governments in malaria control over the past decade, including improved access to health care and malaria rapid diagnostic testing and treatment, clinical malaria incidence remains high [2]. In Mali, in particular, the disease remains highly endemic with resurgence in epidemic prone areas [4]. Many recent studies have shown multiple factors influencing malaria transmission, including socio-economics, demographic and climatic conditions [38,39,40,41] as well as behaviors and access to health care [42]. Environmental, meteorological, vegetation and hydraulic components have been identified to play an important role in malaria transmission dynamics [7,43,44,45]. Since mosquitoes need water to breed, there is a clear relationship with the environment which may modify the dynamics of malaria transmission in specific settings, especially in the Sahelian areas of Africa [46].

It is, therefore, important to study malaria transmission in the context of the global climate change situation [47,48,49,50]. Each epidemiological setting has a particular malaria transmission dynamic associated with its specific environmental factors, population movements, socio-cultural shifts and migration. Hence, the importance of applying geo-epidemiological approaches in malaria is to provide relevant, high-quality evidence to contextualize the complex interrelations among malaria risk factors. In turn, findings from such studies will yield improved adaptive control strategies, mapping solutions, surveillance strategies and targeted interventions. Moreover, the geo-epidemiological approach may play a particularly important role in malaria control in Mali, guiding malaria control efforts toward pre-elimination stage, as acknowledged by the Malian public health authorities [51].

Several studies conducted in West African countries and in Mali [10,11,52] have established the role of meteorological, environmental and hydrologic components in increasing malaria incidence. Some other studies conducted in South Sudan, Ethiopia and in many other areas where malaria is a critical public health concern have established the same influence of climate and environment in malaria incidence increase [53,54,55,56,57,58,59,60]. Most of these studies highlight the role of temperature and humidity associated with higher malaria burdens. However, these components cannot account alone for the rise in malaria cases in these settings. Unlike most of those studies, we assessed the variation of the effects of environment on malaria in different ecosystems in the same time periods. The focus of our study was not only to investigate the nature of the relationship between malaria incidence and environmental, meteorological and hydraulic components in the two locations of Dangassa and Koila, but also to identify components which impact or affect malaria transmission in each of the two Sahelian settings.

The same methodology has been used in our analysis to assure the comparability of the results and to detect site-specific lag duration. Since Dangassa and Koila do not belong to the same ecological zone, it was therefore important to analyze the transmission dynamics in relation to the environment and hydrology, to understand how the incidence of malaria is influenced in each ecosystem.

Our study revealed that meteorological components are significantly associated with malaria transmission in the Sahelian area. In Koila, malaria is more sensitive to meteorological components. Even in the presence of several observed meteorological effects in Dangassa, the influence of the Niger River impacts malaria transmission more than any other environmental factor. We also found that malaria transmission dynamics depends only partly on components such as temperature, vegetation, humidity, and rain in Dangassa. The presence of the Niger River near the village of Dangassa and its regular floods significantly influences the incidence of malaria and modifies the influence of seasonal phenomena on the epidemiology of malaria, thus creating favorable conditions for the development of vectors. The differences in malaria transmission dynamics between the two villages seems to reflect their particularities in terms of geography, environmental setting, population activities, and migrations at certain periods of the year. In fact, our results suggest that the transmission period lasts from July to March in both sites, but compared to Koila, the peak of malaria transmission remains high for a longer period in Dangassa (July to October).

Despite the fact that climate indicators (in average) seemed to be similar in both sites, the time series analysis (Figure 2 and Figure 3) showed that Koila experiences the highest malaria incidence over the study period. In fact, Koila Is located in an irrigate area (Markala Dam on the Niger River), which holds back water flowing downstream from Segou toward Mopti. A second water release occurs in April. This situation favors virtually year-round malaria transmission in Koila driven by irrigation from August to May and by the rainy season from July to November.

The lags observed in Dangassa are quite similar to those observed by Sissoko et al. [10], who found a six week lag between the change in incidence and the Niger River component, compared with seven week lag in our study. At the level of the component consisting mainly of temperature, we obtained a 14 week lag, close to the 13 week lag obtained on a similar principal components analysis study by Sissoko in 2017 in Mali’s peri urban Sotuba site [10]. In Koila, all the meteorological and environmental components were found to be significantly associated with malaria incidence. At sites in West Africa similar to the Koila site [23,35,36], the humidity, vegetation, and rain were found to have an 11 week lag on the change in incidence, which is close to the 12 week lag found by Sissoko in 2017, on the banks of the Niger River [11].

Our results provide information that will contribute to adapting prevention and control policies, rather than giving definitive conclusions in the explanation of the difference, or in the comparison of, the transmission dynamics and the relationship between malaria and environment at the two sites. We believe the findings in this study will serve to optimize the malaria control resources allocation and develop adapted strategies to each particular environmental setting of Mali.

In this study, we used PCA derived variables in our models so they are a result of the combination of multiple variables for axis composition and that could lead to composites cofactors explaining a part one of another of the original variables. Therefore, the multivariate model could express some dominant variable signal among others with respect to their inter relations without losing its pertinence.

In the case of Dangassa the presence of the river (*p* = 0.010) combined with the other variables included in the multivariate model may have blurred the significance of the humidity. Koila did not experienced such a phenomenon.

The effect of the wind speed, higher temperatures and the vegetation on malaria incidence is usually appreciated in a multivariate rather than univariate approach [10,12,61,62,63]. When combined with several other climate and environment factors, the impact of a particular factor on malaria incidence could have shifted from the significance to non-significance and vice versa as observed in Dangassa and Koila on humidity, vegetation, higher temperatures and wind speed. There are some notable limitations for this study. First, it was not possible to assess longer time periods as, to our knowledge, no specific records on the relevant data were recorded prior this study. Further, factors such as personal preventive behaviors and health seeking behaviors may also impact malaria transmission. To better assure the comparability of the two sites, we restricted our study from 22 June 2015 to 31 December 2016, where the two cohorts were followed without interruption. While the same time frame was applied across the two study sites and the environmental data were extracted in the same time span, it was challenging to capture all of the declared malaria cases in the cohort at both sites over the entire study period. We have concentrated our efforts to provide the best estimation of malaria incidence as possible.

Both sites showed some distinct differences in the results of the univariate versus multivariate approaches, particularly those concerning the significance of the main synthetic meteorological factors. These differences can be explained by considering the Niger River as a predictor in the multivariate analysis, which thus became the primary factor in Dangassa. Conversely, for the Koila site, the Niger River was not found to be influential, but synthetic humidity became a key factor influencing the dynamics of malaria transmission.

For the reasons above, the periods of deployment of malaria control interventions should be adjusted to the specific transmission periods of Dangassa, Koila and all sites in similar geographic areas, for a better efficiency in the administration or the control and deployment of such prevention campaigns. In Dangassa, from May to June and in Koila, from April to May, information campaigns, distribution of LLINs, mobilization of community health workers and all other means of public health awareness should be implemented, in order to get the population ready before the beginning of the Seasonal Malaria Chemoprevention (SMC) campaigns, which are carried out just prior to the intense malaria transmission periods in Dangassa and Koila.

## 5. Conclusions

We believe that this geo-epidemiological study has served to elucidate how meteorological conditions (e.g., humidity, temperature) and environmental factors, including river height and vegetation, account for differences in malaria incidence between two ecological settings. The findings here highlight the importance of a geo-epidemiological approach in guiding malaria control interventions with geo-spatial data that may contribute to fine tune the implementation of control strategies.

## Figures and Tables

**Figure 1 ijerph-17-04698-f001:**
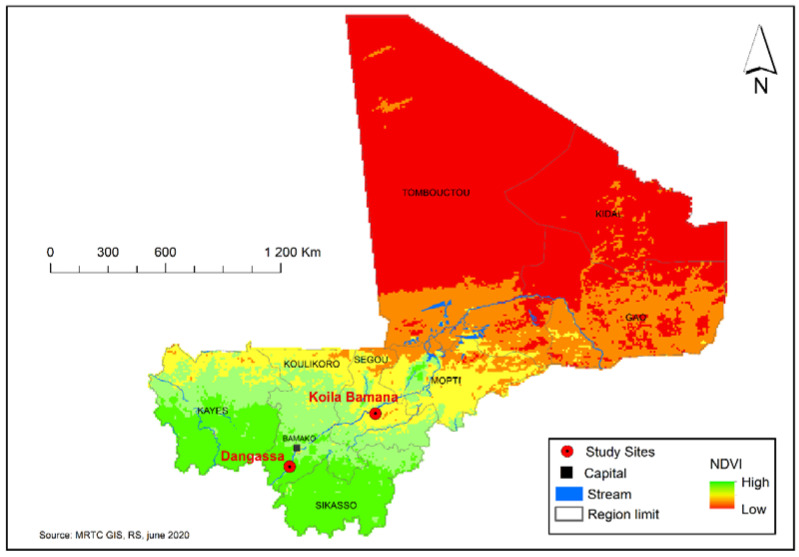
Study sites: shapefiles have been obtained from the Mission of Decentralization and Institutional Reform of Mali (MDRI). The map was based on the cartography of Mali, the mean Normalized Difference Vegetation Index (NDVI) reported downloaded as rasters from the National Aeronautics and Space Administration (NASA) Giovanni on a time period going from 22 June 2015 to 6 January 2017.

**Figure 2 ijerph-17-04698-f002:**
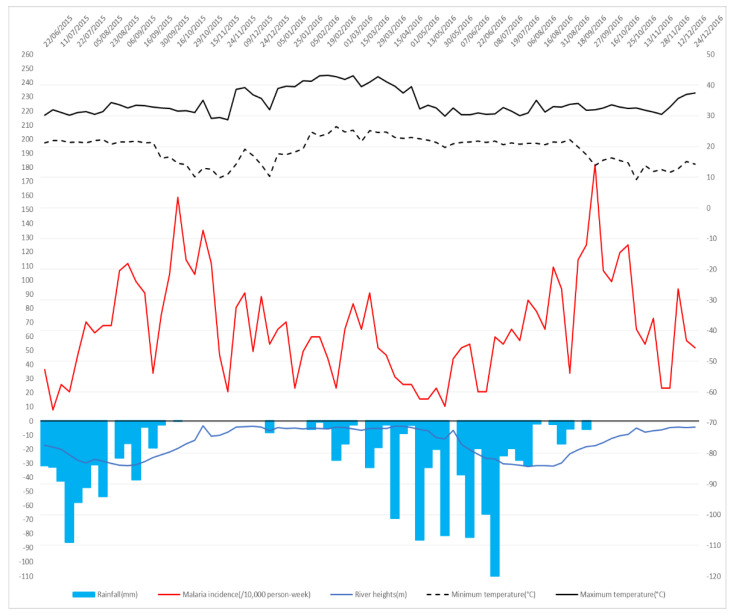
Time series patterns for the Dangassa study site. Malaria incidence time series, maximum temperature, minimum temperature, rainfalls and river height by week. The red line represents weekly malaria incidences, the dashed and solid black lines represent respectively the weekly mean of minimum and maximum air temperature, the solid blue line represents the weekly mean of river height and the blue bar plot represents the weekly cumulative rainfall.

**Figure 3 ijerph-17-04698-f003:**
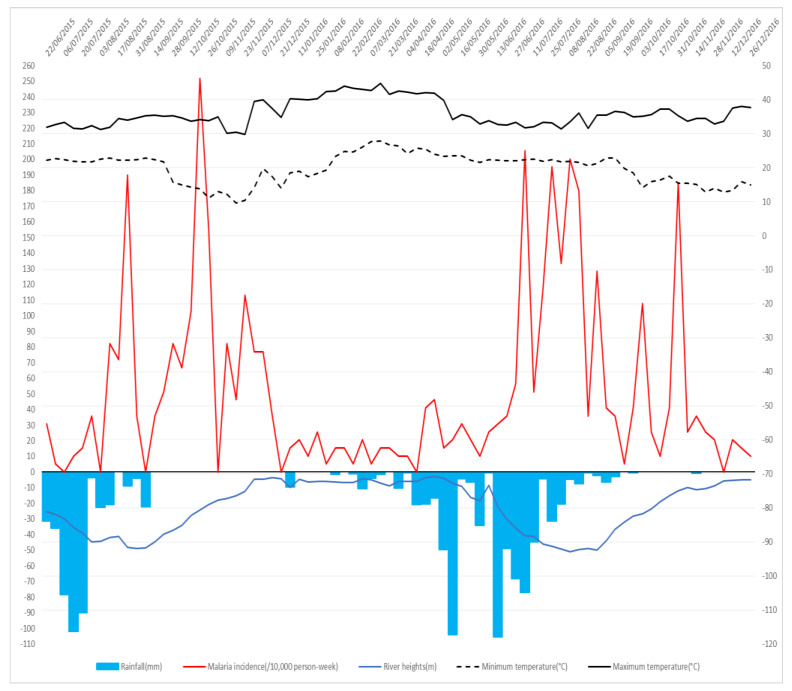
Time series patterns for the Koila study site. Weekly malaria incidence time series, maximum temperature, minimum temperature, rainfalls and river height. The red line represents weekly malaria incidences, the dashed and solid black lines represent respectively the weekly mean of minimum and maximum air temperature, the blue line represents the weekly mean of river height and the blue bar plot represents the weekly cumulative rainfall.

**Figure 4 ijerph-17-04698-f004:**
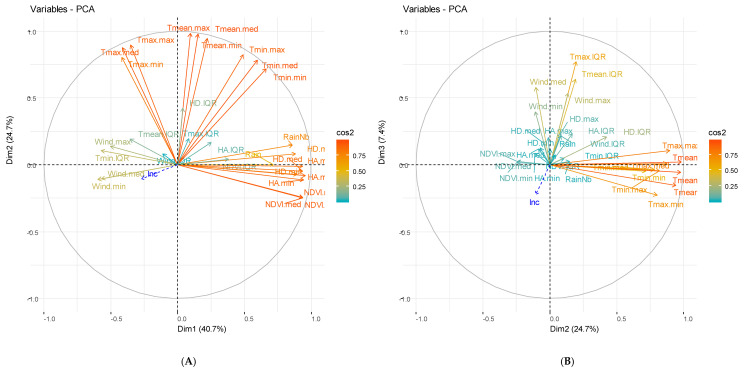
Principal component analysis (PCA) of meteorological and hydrological components for Dangassa study site. (**A**) Represents the two first components for meteorological components, (**B**) the 2nd and 3rd components and (**C**) represents the two first components for hydrological components.

**Figure 5 ijerph-17-04698-f005:**
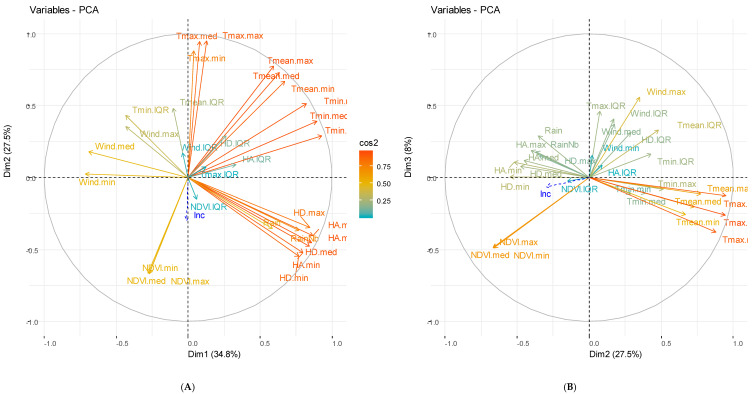
Principal component analysis (PCA) of meteorological and hydrological components for Koila study site. (**A**) Represent the two first components for meteorological components, (**B**) the 2nd and 3rd components and (**C**) represents the two first components for hydrological components.

**Figure 6 ijerph-17-04698-f006:**
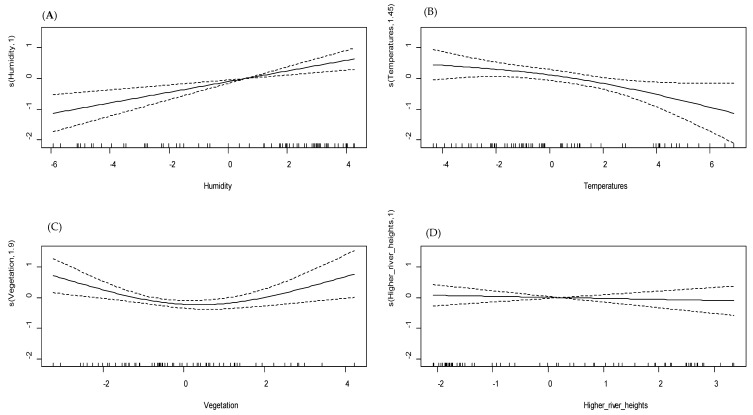
Relationship between meteorological and hydrological components (multivariate GAM model), Dangassa. (**A**) Humidity (Dmt1); (**B**) temperatures (Dmt2); (**C**) wind speed (Dmt3); (**D**) higher river heights (Dhriv1).

**Figure 7 ijerph-17-04698-f007:**
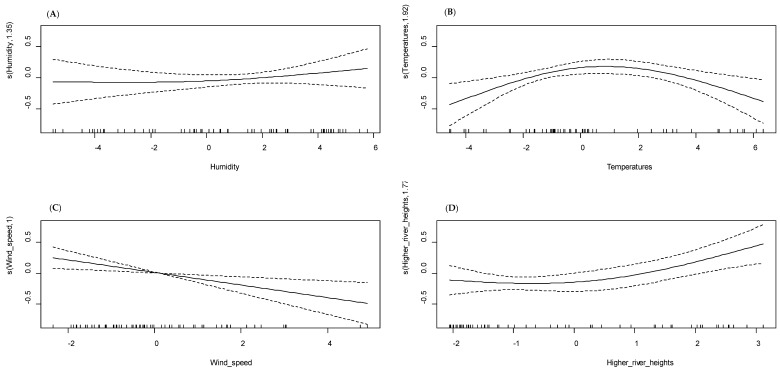
Relationship between climatic, meteorological and hydrological components (multivariate GAM model), Koila. (**A**) Rainfall, temperature, humidity, vegetation, wind speed (Kmt1); (**B**) vegetation, high temperatures (Kmt2); (**C**) all the meteorological components variations (Kmt3); (**D**) high temperatures (Kmt4); humidity, vegetation, wind speed, temperature (Kmt5); river heights (Khriv1).

**Table 1 ijerph-17-04698-t001:** Dangassa and Koila-univariate and multivariate analysis.

Study Site and Characteristic	Univariate Analysis		Multivariate Analysis	
	*p* Value	Dev (%) ^a^	*p* Value	Dev (%) ^a^
Dangassa				
Humidity	<0.0001	34.4	0.54	65
Higher temperatures	<0.001	17.5	0.009	
Wind speed	0.204	11.8	0.005	
High river heights	0.002	22.8	0.01	
Variations in river heights	0.32	1.56		
Koila				
Humidity	0.001	24.8	<0.001	48.2
Higher temperatures	0.205	13.4	0.03	
Vegetation	0.531	4.27	0.004	
High river heights	0.353	10.2	0.66	
Variations in river heights	0.27	10.3		

**^a^** Dev: deviance explained.

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
