# Peer review of "Spatio-Temporal Dynamic of Malaria Incidence: A Comparison of Two Ecological Zones in Mali"

_ijerph, 2020, doi:10.3390/ijerph17134698_

Round 1

Reviewer 1 Report

The authors have demonstrated the relationships between environmental factors and malaria incidence between the two ecological zones in Mali. The experiments and study have been well-designed with good and relatively straightforward results and analysis.

The authors have conveyed a strong message for the adequate malaria control interventions.

It might add more value if the authors can provide a reference to some other similar studies that might have been done in other parts of the world where Malaria is a critical health concern.

Author Response

RESPONSE LETTER

REVIEWER 1:

Comments and suggestions:

It might add more value if the authors can provide a reference to some other similar studies that might have been done in other parts of the world where Malaria is a critical health concern.

Thanks for the review, thank you, we have corrected the manuscript as follow:

“Some other studies conducted in South Sudan, Ethiopia and in many other areas where malaria is a critical public health concern have established the same influence of climate and environment in malaria incidence increase].” (lines 335-337).

References added are (lines 578-593):

  1. Mukhtar, A.Y.A.; Munyakazi, J.B.; Ouifki, R. Assessing the role of climate factors on malaria transmission dynamics in South Sudan. Mathematical Biosciences 2019.
  2. Taddese, A.A.; Baraki, A.G.; Gelaye, K.A. Spatial modeling, prediction and seasonal variation of malaria in northwest Ethiopia. BMC Research Notes 2019.
  3. Le, P.V.V.; Kumar, P.; Ruiz, M.O.; Mbogo, C.; Muturi, E.J. Predicting the direct and indirect impacts of climate change on malaria in coastal Kenya. PLoS ONE 2019.
  4. Tanser, F.C.; Sharp, B.; Le Sueur, D. Potential effect of climate change on malaria transmission in Africa. Lancet 2003.
  5. Rossati, A.; Bargiacchi, O.; Kroumova, V.; Zaramella, M.; Caputo, A.; Garavelli, P.L. Climate, environment and transmission of malaria. Infezioni in Medicina 2016.
  6. Ngarakana-Gwasira, E.T.; Bhunu, C.P.; Masocha, M.; Mashonjowa, E. Assessing the Role of Climate Change in Malaria Transmission in Africa. Malaria Research and Treatment 2016.
  7. Arab, A.; Jackson, M.C.; Kongoli, C. Modelling the effects of weather and climate on malaria distributions in West Africa. Malaria Journal 2014, 13, 126.
  8. Yamana, T.K.; Bomblies, A.; Eltahir, E.A.B. Climate change unlikely to increase malaria burden in West Africa. Nature Climate Change 2016.

Reviewer 2 Report

The study is valuable for understanding the transmission of the malaria and improving the prevention strategies.

Some major points need to be clarified and improved before publication:

  1. Conflicting information of the study time in the abstract and the text, please clarify it.It is 2016-2017 in abstract but 2015 – 2016 in the text.
  2. How will the authors interpret the contradicting results of some variables in table 1:  significant  in the single variant GAD but become nonsignificant in the multivariant analyses, or vice versa? An explanation in the results and discussion will help readers to understand the significance correctly.
  3. Wrong figure citations in the text: line 224 did you mean figure 4p3 and figure 5p3? line231, did you mean figures 4 and 5 ?

Minor points:

1. Grammar mistakes and typos are often seen in the manuscript, please read through and correct them. Such as : Line 37, were should be ‘where’. Line 54, delete the ‘rates’, line56 ‘includes’, and others that I did not list.

2. Also add the abbreviations in figure legends.

3. The order of affiliations is not corresponding to the order it appeared. Please correct it.

Author Response

RESPONSE LETTER

REVIEWER 2:

Thanks for the review, thank you

Comments and suggestions:

Some major points need to be clarified and improved before publication

Conflicting information of the study time in the abstract and the text, please clarify it. It is 2016-2017 in abstract but 2015 – 2016 in the text.

#-------------------------This explanation is not included in the main text of the manuscript--------------- #

#----------------------------------------------------------------------------------------------------------------------------------#

We have corrected the manuscript as follow:

“Malaria data was collected through passive case detection at community health facilities of each study site from June 2015 to January .” (line 29)

“The map was based on the cartography of Mali, the mean Normalized Difference Vegetation Index (NDVI) reported downloaded as rasters from the NASA Giovanni on a time period going from 22/06/2015 to 06/01/2017.” (lines 120-122)

#----------------------------------------------------------------------------------------------------------------------------------#

How will the authors interpret the contradicting results of some variables in table 1:  significant in the single variant GAD but become nonsignificant in the multivariant analyses, or vice versa? An explanation in the results and discussion will help readers to understand the significance correctly.

We have corrected the manuscript as follow in the result part:

We noticed a shift in statistical significance for some covariates in both sites as we moved from the univariate to multivariate model. In Koila, the higher temperatures and the vegetation became significant (p=0.004; p=0.030 respectively). In Dangassa, the humidity became non-significant(p=0.540) when the wind speed moved to significance(p=0.005). (Lines 307-310)

We have corrected the manuscript as follow in the discussion part:

“In this study, we used PCA derived variables in our models so they are a result of the combination of multiple variables for axis composition and that could lead to composites cofactors explaining a part one of another of the original variables .Therefore, the multivariate model could express some dominant variable signal among others with respect to their inter relations without losing its pertinence.

In the case of Dangassa the presence of the river(p=0.010) combined with the other variables included in the multivariate model may have blurred the significance of the humidity. Koila did not experienced such a phenomenon.

The effect of the wind speed, higher temperatures and the vegetation on malaria incidence is usually appreciated in a multivariate rather than univariate approach [10,12,61–63]. When combined with several other climate and environment factors, the impact of a particular factor on malaria incidence could have shifted from the significance to non-significance and vice versa as observed in Dangassa and Koila on humidity, vegetation, higher temperatures and wind speed.” (Lines 380-392)

References added are (lines 592-600):

  1. Alemu, A.; Abebe, G.; Tsegaye, W.; Golassa, L. Climatic variables and malaria transmission dynamics in {Jimma} town, {South} {West} {Ethiopia}. Parasit Vectors 2011, 4, 30.
  2. Kim, Y.M.; Park, J.W.; Cheong, H.K. Estimated effect of climatic variables on the transmission of plasmodium vivax malaria in the republic of Korea. Environmental Health Perspectives 2012.
  3. Bi, Y.; Yu, W.; Hu, W.; Lin, H.; Guo, Y.; Zhou, X.N.; Tong, S. Impact of climate variability on Plasmodium vivax and Plasmodium falciparum malaria in Yunnan Province, China. Parasites and Vectors 2013

 Wrong figure citations in the text: line 224 did you mean figure 4p3 and figure 5p3? line231, did you mean figures 4 and 5?

We have corrected the manuscript as follow:

“For Dangassa and Koila, the combination of hydrological components has made it possible to identify two major components derived from the PCA, representing 99.90% and 100% of the inertia for both sites (see Fig.4P3 and Fig.5P3).” (lines 229-231).

“The combination of meteorological and vegetation components has made it possible to identify three major components (derived from the PCA), accounting for 72.80% and 70.30% of total inertia for Dangassa and Koila, respectively (Figures 4 and 5 ).” (line 238).

Minor points:

Grammar mistakes and typos are often seen in the manuscript, please read through and correct them. Such as: Line 37, were should be ‘where’. Line 54, delete the ‘rates’, line56 ‘includes’, and others that I did not list.

We have corrected the manuscript as follow:

“In Dangassa, the wind speed (p=0.005) and river heights (p=0.010) contributed to increasing malaria incidence, in contrast to Koila, where it was humidity (p<0.001) and vegetation (p=0.004).” (line 38).

“A case-fatality rate of 0.65% was recorded in 2018, and a slight decrease (less than 1%) in the case-fatality rate was observed in 2017 (3).” (line 57).

“Malaria control in Mali is primarily based on improving early access to diagnostics and treatment, which includes: i) intermittent preventive treatment in pregnancy (IPTp, launched in 2003)” (line 59).

A deep and complete English correction have been made overall the manuscript.

Also added the abbreviations in figure legends.

We have corrected the manuscript as follow:

“Figure 4. Principal Component’s analysis (PCA) of meteorological and hydrological components for Dangassa study site.” (line 249).

“Figure 5. Principal component analysis (PCA) of meteorological and hydrological components for Koila study site. “(line 259).

Abbreviations table updated (line 406)

ACTs

Artemisinin-based Combination Therapies

EOSDIS

Earth Observing System Data and Information System

IPTp

Intermittent Preventive Treatment during pregnancy

LLINS

Long Lasting Insecticidal Nets

NASA

National Aeronautics and Space Administration

NDVI

Normalized difference vegetation index

PCA

Principal Components Analysis

PCD

Passive Detection Case

RDT

Rapid diagnostic test

 The order of affiliations is not corresponding to the order it appeared. Please correct it.

We have corrected the manuscript as follow:

“François F. Ateba1,2, Issaka Sagara1, Nafomon Sogoba1, Mahamoudou Touré1, Drissa Konaté1, Sory I. Diawara1, Seidina A. S. Diakité1, Ayouba Diarra1, Mamadou D. Coulibaly1, Mathias Dolo1, Amagana Dolo1, Aissata Sacko1, Sidibe M'Baye Thiam1, Aliou Sissako1, Lansana Sangaré1, Mahamadou Diakité1, Ousmane A. Koita1, Mady Cissoko1,3, Sékou F. Traore1,Peter J. Winch4, Manuel Febrero-Bande5, Jeffrey G. Shaffer6, Donald J. Krogtad2, Hannah C. Marker4 , Seydou Doumbia1* and Jean Gaudart1,3*.

1Malaria Research and Training Center & Department of Public Health Education and Research of the Faculty of Medicine and Odonto-Stomatology, University of Sciences, Techniques and Technologies of Bamako, BP 1805 Point G Bamako, Mali.

2Department of Mathematics, University of Quebec at Montreal (UQAM), 201 President-Kennedy Av., Montréal, QC H2X 3Y7 Canada

3Aix Marseille Univ, INSERM, IRD, SESSTIM UMR1252, Faculty of Medicine, 13005 Marseille, France APHM, La Timone Hospital, Biostatistics & ICT, 13005 Marseille, France.

4Johns Hopkins Bloomberg School of Public Health 615 N. Wolfe Street, Baltimore, MD 21205.

5Dpto. de Estatística, Análise Matemática e OptimizaciónÁrea de Estadística e Investigación Operativa Facultad de Matemáticas Universidad de Santiago de Compostela Campus Vida. 15782 Santiago de Compostela. Spain.

6Department of Global Biostatistics and Data Science, School of Public Health and Tropical Medicine, Tulane University, New Orleans, Louisiana, United States of America, 1440 Canal Street New Orleans, LA 70112

*Emails: [email protected], [email protected].”

(lines  4-22)

Reviewer 3 Report

This study will definitely give an idea of the implementation of the Malaria Eradication Program in the context of environment. I have some following suggestions to the author:

  • Author should correct the captions of figure 2 & 3, e.g. author written in the caption  green, orange line but i can't see that color.
  • Author should explain why malaria incidence is higher in Koila as compare to Dangassa though climatic trend is nearly same in both the sites.

Author Response

RESPONSE LETTER

REVIEWER 3:

Thanks for the review, thank you

Comments and suggestions:

Author should correct the captions of figure 2 & 3, e.g. author written in the caption green, orange line but i can't see that color.

We have corrected the manuscript as follow:

“Figure 2. Time series patterns for the Dangassa study site.  Malaria incidence time series, maximum temperature, minimum temperature, rainfalls and river height by week. The red line represents weekly malaria incidences, the dashedgreen and solid blackpurple lines represent respectively the weekly mean of minimum and maximum air temperature, the solid blueorange line represents the weekly mean of river height and the bluegreen bar plot represents the weekly cumulative rainfall.” (line 209-213).

“Figure 3. Time series patterns for the Koila study site. Weekly malaria incidence time series, maximum temperature, minimum temperature, rainfalls and river height. The red line represents weekly malaria incidences, the dashed and solid black lines represent respectively the weekly mean of minimum and maximum air temperature, the blue line represents the weekly mean of river height and the blue bar plot represents the weekly cumulative rainfall.” (line 222-226).

Author should explain why malaria incidence is higher in Koila as compare to Dangassa though climatic trend is nearly same in both the sites

We have corrected the manuscript as follow:

Despite the fact that climate indicators (in average) seemed to be similar in both sites, the time series analysis (Figure 2 and Figure 3) showed that Koila experiences the highest malaria incidence over the study period. In fact, Koila Is located in an irrigate area (Markala Dam on the Niger River), which holds back water flowing downstream from Segou toward Mopti. A second water release occurs in April. This situation favors virtually year-round malaria transmission in Koila driven by irrigation from August to May and by the rainy season from July to November. (line 366-371).
